

# SILLi 1.0: A 1D Numerical Tool Quantifying the Thermal Effects of Sill
# Intrusions
*Karthik Iyer[1,2], Henrik Svensen[3] and Daniel W. Schmid[1,4]
*karthik.iyer@geomodsol.com
[1] GeoModelling Solutions GmbH, Zurich, Switzerland
[2] GEOMAR, Helmholtz Centre for Ocean Research, Kiel, Germany
[3] Centre for Earth Evolution and Dynamics, University of Oslo, Norway
[4] Physics of Geological Processes, University of Oslo, Norway
## Abstract
Igneous intrusions in sedimentary basins may have a profound effect on the thermal structure and physical
properties of the hosting sedimentary rocks. These include mechanical effects such as deformation and uplift of
sedimentary layers, generation of overpressure, mineral reactions and porosity evolution, and fracturing and
vent formation following devolatilization reactions and the generation of $CO_2$ and $CH_4$. The gas generation and
subsequent migration and venting may have contributed to several of the past climatic changes such as the
end-Permian event and the Paleocene-Eocene Thermal Maximum. Additionally, the generation and expulsion of
hydrocarbons and cracking of pre-existing oil reservoirs around a hot magmatic intrusion is of significant
interest to the energy industry. In this paper, we present a user-friendly 1D FEM based tool, SILLi, which
calculates the thermal effects of sill intrusions on the enclosing sedimentary stratigraphy. The model is
accompanied by three case studies of sills emplaced in two different sedimentary basins, the Karoo Basin in
South Africa and the Vøring Basin offshore Norway. Input data for the model is the present-day well log or
sedimentary column with an Excel input file and includes rock parameters such as thermal conductivity, total
organic carbon (TOC) content, porosity, and latent heats. The model accounts for sedimentation and burial
based on a rate calculated by the sedimentary layer thickness and age. Erosion of the sedimentary column is



also included to account for realistic basin evolution. Multiple sills can be emplaced within the system with varying ages. The emplacement of a sill occurs instantaneously. The model can be applied to volcanic sedimentary basins occurring globally. The model output includes the thermal evolution of the sedimentary column through time, and the changes that take place following sill emplacement such as TOC changes, thermal maturity, and the amount of organic and carbonate-derived $CO_2$. The TOC and vitrinite results can be readily benchmarked within the tool to present-day values measured within the sedimentary column. This allows the user to determine the conditions required to obtain results that match observables and leads to a better understanding of metamorphic processes in sedimentary basins.

# 1    Introduction

Volcanic processes can strongly influence the development of sedimentary basins associated with continental margins. Magmatic bodies such as dikes and sills have a major impact on the thermal evolution of these sedimentary basins. The short-term effects of igneous intrusions include deformation and uplift of the intruded sediments, heating of the host rock, mineral reactions, generation of petroleum, boiling of pore fluids and possible hydrothermal venting (Jamtveit et al., 2004; Malthe-Sorenssen et al., 2004; Svensen et al., 2004; Wang et al., 2012b). Long-term effects include focused fluid flow, migration of hydrothermal and petroleum products, formation of mechanically strong dolerite and hornfels in the contact aureole and differential compaction (Iyer et al., 2013; Iyer et al., 2017; Kjoberg et al., 2017; Planke et al., 2005). This is of particular importance to understanding the carbon cycle, as thermal stresses, besides those associated with burial, encountered by organic matter in immature source rocks will determine the ultimate production and fate of the $CO_2$ and $CH_4$ generated. Vent structures are intimately associated with sill intrusions in sedimentary basins globally and are thought to have been formed contemporaneously due to overpressure generated by pore-fluid boiling gas generation during thermogenic breakdown of kerogen (Aarnes et al., 2015; Iyer et al., 2017; Jamtveit et al., 2004). Methane and other gases generated during this process may have driven catastrophic climate change in the geological past (Svensen and Jamtveit, 2010; Svensen et al., 2009). In order to understand these problems, numerical models are widely used to reconstruct the thermal history of a basin where only a few of these parameters are known.



A number of analytical and numerical models have been developed that study the thermal effects of igneous intrusions dating back to the early- and mid-1900's (Jaeger, 1964; Jaeger, 1957, 1959; Lovering, 1935). Subsequent 1D and 2D models added additional complexity to the models by the addition of emplacement mechanisms and timing, source rock maturation, hydrocarbon generation, latent heats of devolatilization and maturation, fluid processes and overpressure generation (Aarnes et al., 2011a; Fjeldskaar et al., 2008; Galushkin, 1997; Iyer et al., 2017; Monreal et al., 2009; Wang, 2012; Wang et al., 2010; Wang and Song, 2012; Wang et al., 2012a). Contact metamorphic processes are well understood (e.g. (Aarnes et al., 2010; Jamtveit et al., 1992; Tracy and Frost, 1991)), but many published papers do not take into account the basin history or the variations in contact aureole thickness that arise from the type of measuring method that has been used. In general, the contact metamorphic effects depend on 1) sill thickness (note that dikes cannot be directly compared with sills), 2) sill emplacement temperature, 3) thermal gradient and emplacement depth (i.e. temperature and background maturation), 4) emplacement history (instantaneous versus prolonged magma flow), 5) host rock composition and characteristics (such as thermal conductivity, organic carbon content, porosity, permeability) and 6) conductive versus advective cooling (e.g. (Aarnes et al., 2010; Galushkin, 1997; Iyer et al., 2013; Iyer et al., 2017; Jaeger, 1964; Lovering, 1935; Wang, 2012)). In addition, the contact aureole width depends on how aureoles are studied and measured. The aureole thickness depends on the proxy used, including sonic velocity, density, mineralogy and mineral properties, magnetic susceptibility, total organic carbon content, vitrinite reflectivity, color, porosity, or organic geochemistry. Note that these aureole thickness proxies will not necessarily give the same result. Finally, the aureole thickness also depends on the proximity to other sills emplaced at the same time (see Aarnes et al. (2011b) for a quantification).

In this paper we present a generic 1D thermal model, SILLi, which can be applied to studying the thermal effects of sill intrusions in sedimentary basins globally. Besides heat transfer, the model also accounts for the sequential deposition of sedimentary layers through time, erosion, latent heat effects and gas generation by decarbonation reactions and organic matter maturation. The model results can be then easily compared to the two most widely used aureole proxies in sedimentary rocks, vitrinite reflectance (VR) and total organic carbon (TOC) data.



## 2    Model Input

The one dimensional, Finite Element Method (FEM) model numerically recreates the thermal effects of sill emplacement in a sedimentary column. The model is written using MATLAB and requires version 2014b or higher to run. The model input is specified in an Excel (*.xls) file and is read by the Matlab file, SILLi.m. The user also specifies the model resolution with the igneous intrusions and sedimentary layers by giving the minimum spacing (m) or the minimum number of points in the Matlab file. The measure that produces the highest resolution is used. The Excel file is composed of seven tabs outlined below. If a previously calculated output file is available for the input file, the program prompts the user to choose between loading the output file for further analysis and performing a new calculation which overwrites the existing file.

For correct model use, the geological input needs to be based on either a borehole (with horizontal stratigraphy) or an outcrop that is converted into a pseudo-borehole. If the case study is outcrop-based, a pseudo-borehole stratigraphy should be constructed including the regional basin stratigraphy. Note that sedimentary rocks present at higher stratigraphic levels elsewhere in the basin should be added to the erosion history of the basin. Moreover, the sills (and samples) should be rotated back to horizontal if the stratigraphy was tilted post sill emplacement. Using TOC and VR data from sedimentary rocks outside the immediate contact aureoles will improve the model calibration.

### 2.1   Fluid

This tab contains three columns describing the fluid name, its density (kg/m$^3$) and its heat capacity (J/kg/K).

### 2.2   Lithology

This tab contains the data required for the model to build the present-day sedimentary column. The various columns detail the name of the sedimentary layer (character only) and various material properties such as density (kg/m$^3$), heat capacity (J/kg/K), porosity (fraction), thermal conductivity (W/m/K), initial TOC content (wt%) and latent heats of organic maturation and dehydration (kJ/kg). Information regarding the kind of carbonate contained in the sedimentary layer can be given in the last column if decarbonation reactions are



considered. The mineral constitution of the carbonate can be chosen as marl (1), dolomite (2) or
dolomite/evaporite mix (3). A zero (0) is entered in this column if decarbonation reactions are not required. The
lithology tab also contains columns where the present-day top depth (m) and age (Ma) of each layer can be
given which determine the depositional sequence and sedimentation rate for the layer (see Section 3.1). Note
that the ages of the sedimentary must be unique. A hypothetical basement is added 10 m below the deepest
sedimentary layer top depth or 300 m below the bottom of the deepest sill intrusion, whichever is deeper.

## 2.3   Erosion

This tab is similar to the lithology tab and contains information on eroded layers. Additional columns in this tab
contain information regarding the erosion timing (Ma) and the thickness of the eroded layer (m). Note that the
top depth of the eroded layer must coincide with the top of a sedimentary layer in the lithology tab. If part of
sedimentary layer is indeed eroded before deposition continues (i.e. the eroded layer lay inside a deposited
layer), the layer needs to be considered as unique layers separated by the eroded layer. Multiple eroded layers
can have the same top depths provided that older layers with the same top depth are eroded first. Similarly,
eroded layers have to be eroded first prior to deposition of younger layers. The ages of the eroded layers
cannot coincide with other layers.

## 2.4   Sills

This tab contains information necessary for the emplacement of sill intrusions. The top depth (m) and thickness
(m) of the sill constrain the geometry of the intrusion. Additional information includes the time of emplacement
(Ma), emplacement temperature (°C), melt and solid densities (kg/m$^3$), melt and solid heat capacities (J/kg/K),
thermal conductivity (W/m/K), solidus and liquidus temperatures of the magma (°C) and the latent heat of
crystallization (kJ/kg). The emplacement of the intrusion is assumed to be instantaneous. Note that the top
depth of the sill cannot be the same as the top depth of a sedimentary layer. On the same note, the top depth
of a sedimentary layer cannot be inside a sill intrusion. Emplacement ages cannot exactly coincide with layer
ages.

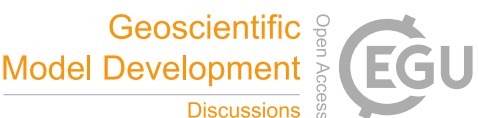


## 2.5 Temperature Data

This tab contains temperature data (°C) vs. depth (m) for the sedimentary column. The data in this tab is used to construct a geothermal gradient by using the best linear fit and therefore needs to contain at least two data points. Additionally, the first data point must coincide with the column top describing the surface temperature.

136

## 2.6 Vitrinite Data (Optional)

This tab contains present day vitrinite reflectance data presented in depth (m) and VR values (%Ro). Standard deviation of the values when available can be included. This data is used for comparison of the modelled VR values to observations. This tab can be left blank if no information is available.

141

## 2.7 TOC Data (Optional)

This tab contains present day TOC content data (wt%) vs. depth (m) measured in the sedimentary column which is used to compare to the model results. This tab can be left blank if no information is available.

145

# 3 Method

## 3.1 Sediment Deposition and Erosion

Each sedimentary layer, including the eroded layers, is deposited sequentially in time based on the depositional age. The rate of sedimentation for each layer is determined by the thickness of the layer and the difference in time between its top age and that of the layer deposited before it. Erosional layers in the sedimentary column are deposited in the same way as other layers. Erosion of the entire layer occurs within a single step at the specified erosion age. The temperature boundary conditions are accordingly adjusted for the height of the new sedimentary column. Note that the bottom boundary is extended to 5 times the thickness of the bottommost sill if that sill is close to or at the bottom boundary in order to remove boundary effects.



155

## 3.2 Thermal Diffusion

The thermal solver computes the temperature within the deposited sedimentary column by applying fixed temperatures at the top and bottom at every step which are calculated from the prescribed geotherm (see Section 2.5) and the energy diffusion equation,

$$\left[ \phi \rho_f c_{pf} + (1-\phi) \rho_r c_{peff} \right] \frac{\partial T}{\partial t} = \nabla \cdot (\kappa \nabla T) \qquad (1)$$

Table 1 contains the definitions of all the notations used in the manuscript. The effective rock heat capacity accounts for the latent heat of fusion in the crystallizing parts of the sill between the solidus ($T_S$) and liquidus ($T_L$) temperature of the magma (e.g. (Galushkin, 1997))

$$
\begin{aligned}
c_{peff} &= c_{pm} \left[ 1 + \frac{L_c}{(T_L - T_S) c_{pm}} \right] \text{ if } \left[ T_S < T < T_L \right] \\
c_{peff} &= c_{pr} \qquad\qquad\qquad \text{ if } \left[ T_S > T \right]
\end{aligned}
\qquad (2)
$$

Sills are emplaced instantaneously at the specified time and temperature within the sedimentary column. The emplacement of multiple sills in the same step is possible. The time-steps used for thermal diffusion after sill emplacement are automatically calculated based on the sill thickness and the characteristic time required for thermal diffusion. The time step is initially small in order to accurately resolve the thermal evolution of the contact aureole around the sill and is gradually increased once the energy released by the cooling sill is dissipated.

Dehydration reactions in the host rock are implemented by modifying the thermal diffusion equation when temperatures of the sediments increase within a certain range (Galushkin, 1997; Wang, 2012)

$$\left[ \phi \rho_f c_{pf} + (1-\phi) \rho_r c_{peff} \right] \frac{\partial T}{\partial t} = \nabla \cdot (\kappa \nabla T) - H \qquad (3)$$



174

$$H = \frac{(1-\varphi)\,\rho_r L_d}{T_{d1} - T_{d2}} \frac{\partial T}{\partial t}$$

(4)

| Symbol | Description | Units |
|:---:|:---:|:---:|
| $A$ | Frequency factor | $s^{-1}$ |
| $c_{peff}$ | Effective rock heat capacity | $J\ kg^{-1}\ K^{-1}$ |
| $c_{pf}$ | Fluid heat capacity | $J\ kg^{-1}\ K^{-1}$ |
| $c_{pr}$ | Rock heat capacity | $J\ kg^{-1}\ K^{-1}$ |
| $E$ | Activation energy | $KJ\ mol^{-1}$ |
| $f$ | Stoichiometric factor | |
| $F$ | Reaction extent | |
| $g$ | Gravitational acceleration | $m\ s^{-2}$ |
| $i$ | Reactive component | |
| $L_c$ | Latent heat of crystallization | $KJ\ kg^{-1}$ |
| $m_{CO_2}$ | Carbon to $CO_2$ conversion factor | 3.66 |
| $P_{atm}$ | Atmospheric pressure | $10^5$ Pa |
| $P_{H2O}$ | Hydrostatic pressure | Pa |
| $R_{CO_2}$ | Rate of $CO_2$ generation | $kg\ m^{-3}\ s^{-1}$ |
| $R_{om}$ | Rate of organic matter degradation | $kg\ m^{-3}\ s^{-1}$ |
| $t$ | Time | s |
| $T_L$ | Liquidus temperature | °C |
| $T_S$ | Solidus temperature | °C |
| $T$ | Temperature | °C |
| $T_{d2}$ - $T_{d1}$ | Temperature range for dehydration reactions (Galushkin, 1997) | 350-650 °C |
| $w$ | Amount of reactive component | Fraction |
| $Z$ | Depth | km |
| $\phi$ | Rock porosity | Fraction |





| $\kappa$ | Conductivity | W m$^{-1}$ K$^{-1}$ |
|---|---|---|
| $\rho_f$ | Fluid density | kg m$^{-3}$ |
| $\rho_r$ | Rock density | kg m$^{-3}$ |

Table 1. Definition of symbols used in the model.

## 3.3    Thermal Maturation of Organic Matter

Vitrinite reflectance is a widely used indicator of thermal maturity and can be readily measured in the field. One of the most common methods used to calculate the thermal maturity of the source rock is the EASY%Ro method put forward by Sweeney and Burnham (1990). This model uses 20 parallel Arrhenius-type of first order reactions to describe the complex process of kerogen breakdown due to temperature increase. The reaction for the $i^{th}$ component is given by

$$\frac{dw_i}{dt} = -w_i A \exp\left[-\frac{E_i}{RT^t}\right] \tag{5}$$

where $w_i$ is the amount of material for component $i$, $E_i$ is the activation energy for the given reaction and $T^t$ is time-dependent temperature.

The total amount of material reacted is obtained by summing up the individual reactions

$$\frac{dw}{dt} = \sum_i \frac{dw_i}{dt} \tag{6}$$

The fraction of reactant converted is

$$F = 1 - \frac{w}{w_0} = 1 - \sum_i f_i\left(\frac{w_i}{w_{0i}}\right) \tag{7}$$

from which the vitrinite reflectance can be readily calculated by





$$\% Ro = \exp\left(-1.6 + 3.7F\right) \qquad (8)$$

The amount of TOC that has reacted for any given time can be calculated by

$$\text{TOC}(t) = \text{TOC}_o F(t) \qquad (9)$$

and the rate of organic matter degradation by

$$R_{om} = \left(1 - \phi\right)\rho_r \frac{\partial \text{TOC}}{\partial t} \qquad (10)$$

The maximum amount of TOC that can be reacted by this method is 85% of the initial total. Note that in the inner part of the contact aureole close the sill, data shows that all of the organic matter has been reacted or removed (eg. LA1/68 in section 5.2.2). We assume that all of the hydrocarbons released during thermal degradation are converted into carbon dioxide. The amount of organic carbon dioxide generated ($R_{CO2}$) for a time step is given by

$$R_{CO_2} = R_{om} m_{CO_2} \qquad (11)$$

where $m_{CO2}$ is a stoichiometric conversion factor (3.67) to transform carbon into carbon dioxide. Note that metamorphism of sedimentary rocks will generate $CH_4$ (e.g., (Aarnes et al., 2010; Iyer et al., 2017)), but in our model the reacted carbon is recalculated to $CO_2$. If needed, the $CO_2$ model output can be easily converted to either C or $CH_4$.

The latent heat of organic maturation is accounted for in the energy equation

$$\left[\phi\rho_f c_{pf} + \left(1 - \phi\right)\rho_r c_{peff}\right]\frac{\partial T}{\partial t} = \nabla \cdot \left(\kappa \nabla T\right) - H - L_{om} R_{om} \qquad (12)$$

## 3.4  Mineral Decarbonation

Carbonate minerals undergo decarbonation reactions as they are heated to high temperatures. This results in mineral transformations and the release of inorganic carbon dioxide which may significantly add to the $CO_2$



budget associated with igneous intrusions. The amount of inorganic $CO_2$ liberated during metamorphic
transformation over a range of temperature and fluid pressure for marl, dolomite and dolomite/evaporite
mixture is pre-computed as a phase diagram using Perple_X (Connolly and Petrini, 2002) (Figure 1). The model
evaluates the total amount of inorganic $CO_2$ liberated by carbonate layers based on the temperature and
pressure evolution of the layer through time within the phase diagrams. Fluid pressure within the sedimentary
column is calculated by integrating the rock density over depth in addition to atmospheric pressure:
$$P_{H_2O} = P_{atm} + \int_{Z_{min}}^{Z_{max}} \rho_f \vec{g} \qquad (13)$$

**A** Free $CO_2$ in Marl [wt%]

**B** Free $CO_2$ in Dolostone [wt%]

**C** Free $CO_2$ in Dolostone/Evaporite [wt%]




*Figure 1. Phase diagrams generated by Perple_X showing the amounts of inorganic $CO_2$ liberated with respect to*
*temperature and pressure for marl (A), dolostone (B) and dolostone/evaporite (C).*

## 3.5  Model Mesh and Time-Stepping

The entire sedimentary column including the eroded layers and igneous intrusions is reconstructed and the
column nodes and elements for the FEM model are generated using the user-specified resolution. The nodes
are initially collapsed onto each other in depth. Each sedimentary node is assigned a time during which it is
expanded (or deposited) within the sedimentary column based on the layer age and its thickness. All of the
elements and nodes associated with each igneous intrusion are expanded simultaneously during the
corresponding emplacement time. Eroded layers are removed in a single time step specified by the erosion age
and the corresponding nodes are collapsed. In order to correctly capture thermal diffusion across the large
thermal gradient adjacent to a hot intrusion, the time step is initially very small and exponentially increases
during the heating period after sill emplacement and before the next depositional event. The heating period of
the sill, over which the exponential time sub-stepping is used, is analytically determined from the characteristic
diffusion time for the sill thickness (Jaeger, 1959).

## 3.6  Model Limitations

• The model is one-dimensional and will therefore not resolve thermal effects that would require a full 3D
model.
• The model does not account for advective transport of heat through the system by fluids. However,
previous models have shown that this process would be dominant only in high permeability systems or
at the sill edges/tips in low permeability systems (Iyer et al., 2013; Iyer et al., 2017). Therefore, the
model presented in this manuscript works well for relatively low permeability systems with shales,
mudstone etc. and when the sedimentary column passes through the sill interior away from the edges.
• The model does not account for other mineral reactions in the contact aureole besides decarbonation
of carbonates. The various mineral reactions possible in the contact aureole can be implemented as an
add-on module to the model if needed.
• The model assumes that TOC conversion in all types of sedimentary rocks can be estimated by using the
EASY%Ro method with a maximum conversion value of 85%. Although, this is a good first
approximation, it cannot account for the complete loss of carbon in zones very close to the sill-host rock
interface which would result in an underestimation of the released gases (Svensen et al., 2015). On the



other hand, the provenance of the sedimentary rock can also significantly affect how kerogen present in
organic matter reacts to form hydrocarbons which may result in a reduction in the amount of
convertible organic matter due to the presence of inert kerogen (Iyer et al., 2017; Pepper and Corvi,
253    1995).


# 255    **4    Model Output**

The model input and results are presented with the help of a GUI (Section 4.6). Model data are written out as a
single .mat (Matlab data) file in the same directory as the user-defined path for the input Excel file and with the
same filename. The file contains five 'struct' variables of which three contain input information (rock, sill and
welldata) and the other two contain model results (result and release). The structure of the variables are
described below.

## 262    **4.1   Struct Variable: rock**

This variable contains input information on the sedimentary layers in the column including the eroded layers.
The information is saved as variables given in Table 2 and is sorted according to their top depths. Note that top
depths are corrected for the eroded layers that are also included.

| Variable Name | Description |
|---|---|
| Name | User-defined names of all the sedimentary layers in the column. |
| num | Total number of deposited sedimentary layers. |
| top | Top depth of the shallowest sedimentary layer. |
| bot | Top depth of the deepest sedimentary layer. |
| Tops | Top depths of sedimentary layers. |
| Ages | Ages of sedimentary layers. |
| Rho | Density of sedimentary layers. |
| Cp | Heat capacity of sedimentary layers. |
| Phi | Porosity of sedimentary layers. |



| K | Thermal conductivity of sedimentary layers. |
|---|---|
| Toc | TOC content of sedimentary layers. |
| Lm | Latent heat of maturation of sedimentary layers. |
| Ld | Latent heat of dehydration of sedimentary layers. |
| Carb | Carbonate layer identifier (0-3). |
| Ero_t | Erosion age of sedimentary layers (NaN if layer is not eroded). |
| Ero_thick | Eroded thickness of sedimentary layers (NaN if layer is not eroded). |
| Ero_tops | Top depths of the eroded layers only. |

*Table 2. List of variables in 'rock' struct variable of the output file.*

## 4.2   Struct Variable: sill

This variable contains input information on sill intrusions in the column. The information is saved as variables
given in Table 3 and is sorted according to their top depths.

| Variable Name | Description |
|---|---|
| num | Total number of sill intrusions. |
| Tops | Top depths of sill intrusions. |
| E_time | Emplacement ages of sill intrusions. |
| E_temp | Emplacement temperatures of sill intrusions. |
| Rhom | Melt density of sill intrusions. |
| Cpm | Melt heat capacity of sill intrusions. |
| Rhos | Solid density of sill intrusions. |
| Cps | Solid heat capacity of sill intrusions. |
| K | Thermal conductivity of sedimentary layers. |
| Sol | Solidus of melt in sill intrusions. |
| Liq | Liquidus of melt in sill intrusions. |
| Ld | Latent heat of crystallization of melt in sill intrusions. |



*Table 3. List of variables in 'sill' struct variable of the output file.*

## 4.3   Struct Variable: welldata

This variable contains input information on measured TOC, VR and temperature data for the sedimentary
column. The information is saved as variables given in Table 4.

| Variable Name | Description |
|---|---|
| TOC | Measured TOC data vs. depth. |
| VR | Measured VR data vs. depth. |
| T | Measured temperature data vs. depth. |

*Table 4. List of variables in 'welldata' struct variable of the output file.*

## 4.4   Struct Variable: result

This variable contains the model results which are saved for every time step when applicable, i.e. variables that
change over time have rows corresponding to the element or node number (depending on where they are
defined) and columns corresponding to the time step number. The information is saved as variables given in
Table 5.

| Variable Name | Description (Rows x Columns) |
|---|---|
| nel | Number of elements in the model (1 x 1) |
| nnod | Number of nodes in the model (1 x 1) |
| Gcoord_c | Depth of element centers (1 x no. of elements) |
| Ind | Internal nodal indexing of sedimentary layers and intrusions (no. of nodes x 1). Intrusions are negatively indexed. |
| Ind_nel | Internal element indexing of sedimentary layers and intrusions (no. of elements x 1). Intrusions are negatively indexed. |
| Ind_carb | Nodal indexing of carbonate layers (0-3) (no. of nodes x 1). |





| | |
|---|---|
| **Gcoord** | Depth of nodes (no. of nodes x no. of time steps). |
| **Temp** | Nodal temperature (no. of nodes x no. of time steps). |
| **Pres** | Nodal hydrostatic pressure (no. of nodes x no. of time steps). |
| **Toc** | Remaining Toc content at nodes (no. of nodes x no. of time steps). |
| **CO2_org** | Organic carbon dioxide generated at nodes (no. of nodes x no. of time steps). |
| **Ro** | VR at nodes (no. of nodes x no. of time steps). |
| **Tmax** | Maximum temperature experienced at nodes (no. of nodes x no. of time steps). |
| **Active** | Binary index of 'deposited/expanded' nodes (no. of nodes x no. of time steps). |
| **CO2_release** | Inorganic carbon dioxide generated at nodes (no. of nodes x no. of time steps). |
| **Time** | Year count for time step (no. of time steps x 1). |

Table 5. List of variables in 'result' struct variable of the output file.

## 4.5 Struct Variable: release

This variable contains the amounts of $CO_2$ released for every time step normalized to rock volume. The information is saved as variables given in Table 6.

| **Variable Name** | **Description (Rows x Columns)** |
|---|---|
| **CO2_org** | Organic carbon dioxide generated in elements normalized to rock volume (no. of elements x no. of time steps). |
| **CO2_rel** | Inorganic carbon dioxide generated in elements normalized to rock volume (no. of elements x no. of time steps). |

Table 6. List of variables in 'release' struct variable of the output file.

## 4.6 Output Graphical User Interface (GUI)

The GUI presented during and after the model run contains three tabs containing graphical representations of the input data, time evolution of model results and $CO_2$ release through time. An explanation of the tabs is



given below using a hypothetical test case consisting of a sedimentary column with two sill intrusions and three
eroded layers.

### 4.6.1    Input Tab

The left-most subplot of the input tab contains the reconstructed sedimentary column where the layers are
colored according to their depositional age (http://www.stratigraphy.org/index.php/ics-chart-timescale) (Figure
2). The sedimentary column also contains eroded layers (hatched) and sill intrusions (speckled). The name and
depositional age of a layer can be found by right-clicking the layer. The other subplots in the input tab contain
information on the density, porosity, initial TOC content and thermal conductivity of the sedimentary layers.
The values of these variables are plotted at the corresponding layer top depth.

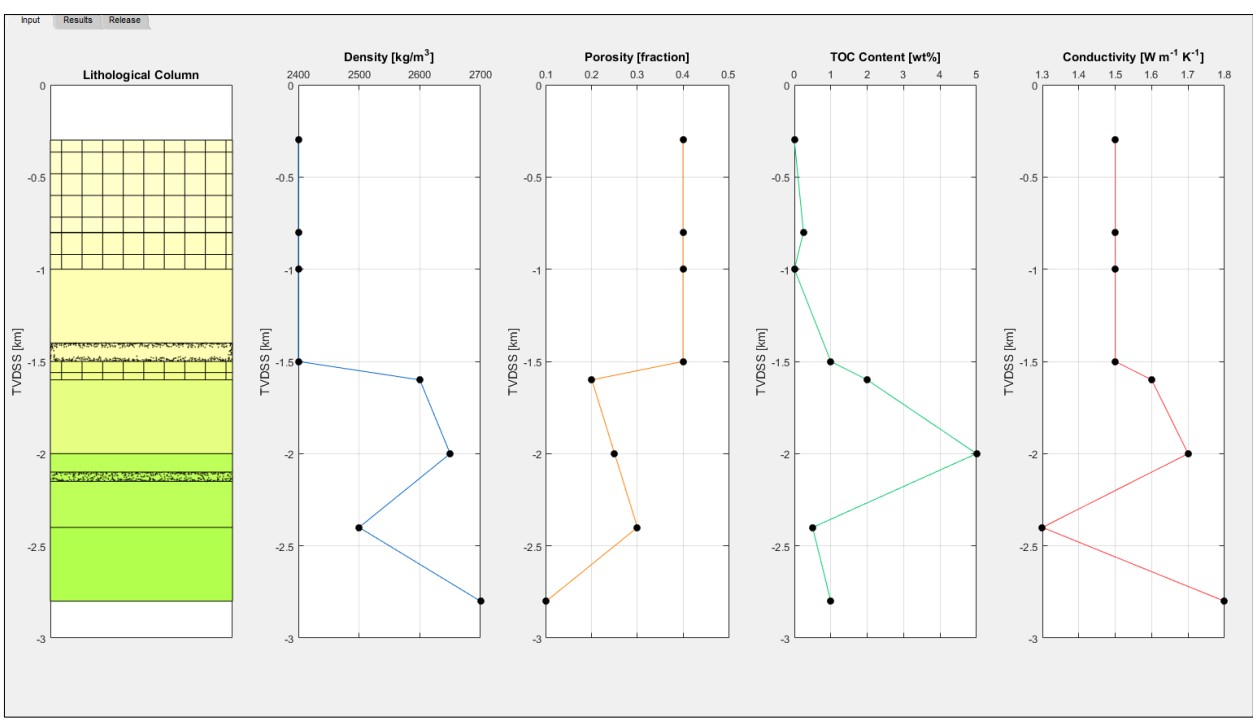


*Figure 2. Snapshot of the input tab generated for a hypothetical sedimentary column with two sill intrusions and three eroded layers. Right-clicking a layer in the sedimentary column provides the name and depositional/erosional age of the layer.*



### 4.6.2 Results Tab

The results tab consists of the evolution of temperature, vitrinite reflectance, TOC content, maximum
temperature, hydrostatic pressure, inorganic and organic $CO_2$ release within the sedimentary column over
simulated time (Figure 3). The evolution of these variables can be played or stepped through using the player
controls in the top left corner. Alternatively, the user can jump directly to the desired geological time by
inputting it in the player control. Note that this results in the plot jumping to the time-step nearest the desired
time input. Regions containing sill intrusions are highlighted in gray. Users can copy plot data at any time step
by right-clicking the curve.
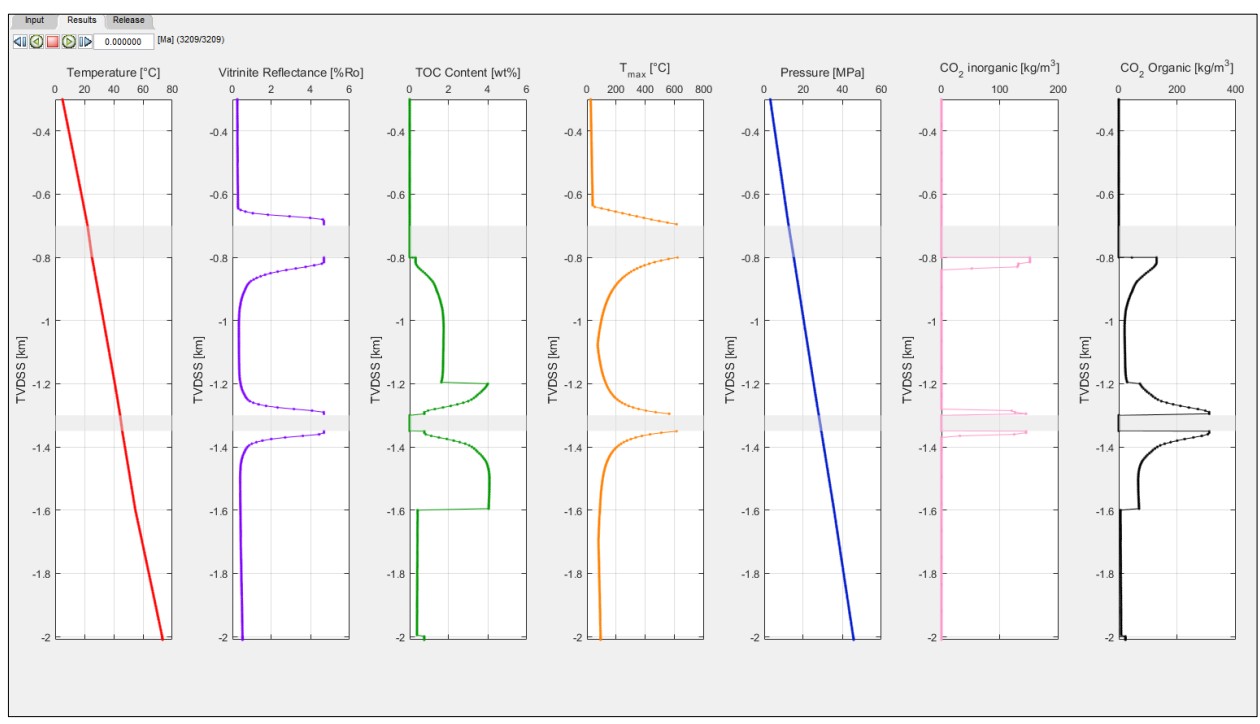
*Figure 3. Snapshot of the results tab generated for a hypothetical sedimentary column with two sill intrusions*
*and three eroded layers. Right-clicking any curve allows the user to copy curve data.*

### 4.6.3 Release Tab

The release tab plots the cumulative and rates of release of organic and inorganic $CO_2$ due to heating of the
sedimentary layer by sill intrusions (Figure 4). The cumulative and release rates are summed over the entire
sedimentary column. The user can use the cumulative amount of gas released to easily upscale to basin scales





by multiplying the value by the area affected by sill intrusions. Users can copy plot data at any time step by
right-clicking the curve.

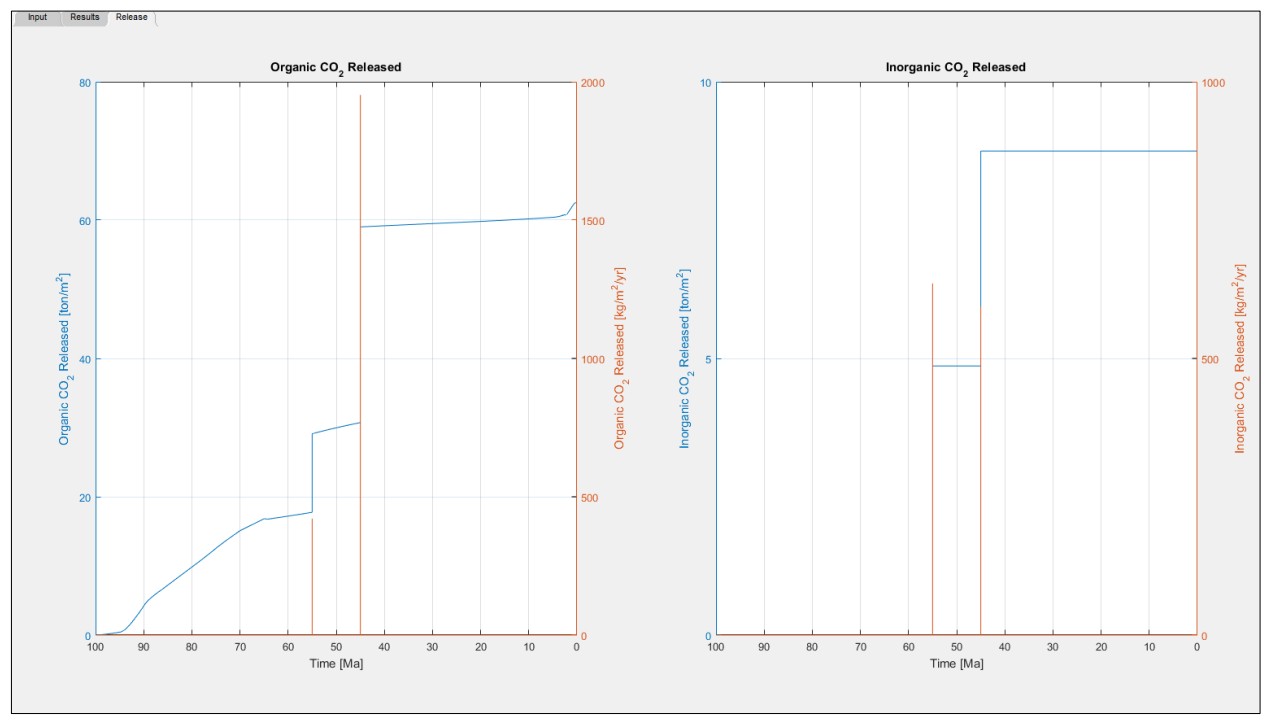

*Figure 4. Snapshot of the release tab generated for a hypothetical sedimentary column with two sill intrusions*

*and three eroded layers. Right-clicking any curve allows the user to copy curve data.*

# 5    Examples

The examples below are provided with the code and are used to benchmark observations to model results.

## 5.1   Utgard High

The Utgard sill complex is part of the North Atlantic Igneous Province (NAIP) in the Vøring and Møre Basins,

offshore Norway. This region underwent massive volcanic activity at the Paleocene-Eocene boundary around

~55 Ma (Aarnes et al., 2015). The Utgard High borehole 6607/5-2 was drilled through two sills emplaced in the

Upper Cretaceous sedimentary layers. The drilled lithological column consists of nine layers with the oldest



being deposited 100 Ma (NPD Factpages, http://factpages.npd.no/factpages/) (Figure 5). For simplicity, the
material properties of the entire sedimentary column is set to constant values with the exception of TOC
content. TOC content of the Paleocene and Upper Cretaceous sedimentary layers are set to an initial value of
0.6 and 1.5 wt%, respectively. Carbonate and erosional layers are not considered. The modelled sedimentary
layers are sequentially deposited at the sedimentation rate calculated from the layer top ages. The two sills are
emplaced simultaneously within the Nise and Kvitnos Formations at 55 Ma at a temperature of 1150°C.
Sedimentary rocks around the emplaced sills are progressively heated as the sills cool. The vitrinite reflectance
values increase and the TOC content reduced by thermally degrading organic matter to form $CO_2$ (Figure 6).
Sedimentation after sill emplacement results in further burial and extension to produce the present-day
sedimentary column. Vitrinite reflectance and TOC data from the Norwegian Petroleum Directorate (NPD) and a
previous study (Aarnes et al., 2015) are used to benchmark the model and match very well with the modelled
results (Figure 7). Further information about the geological and model setting can be found in Aarnes et al.
(2015) and the input file '1d_sill_input_utgard.xlsx'.

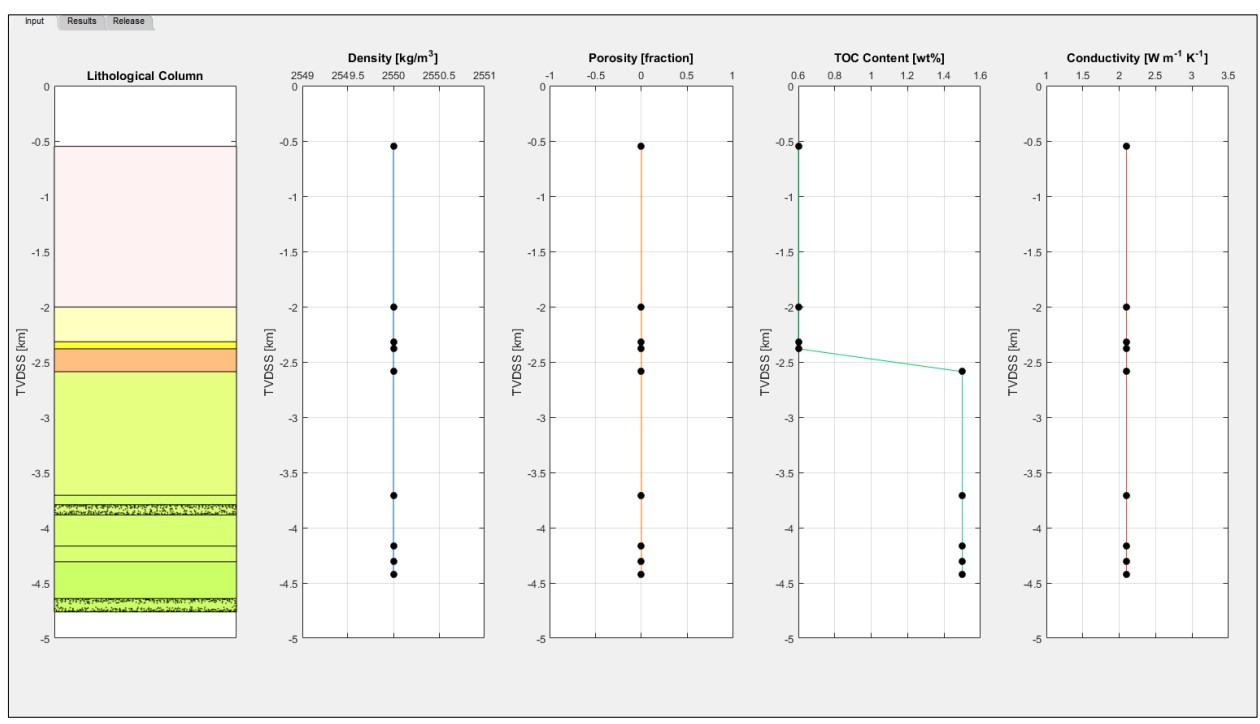


*Figure 5. Input tab for the Utgard High example.*



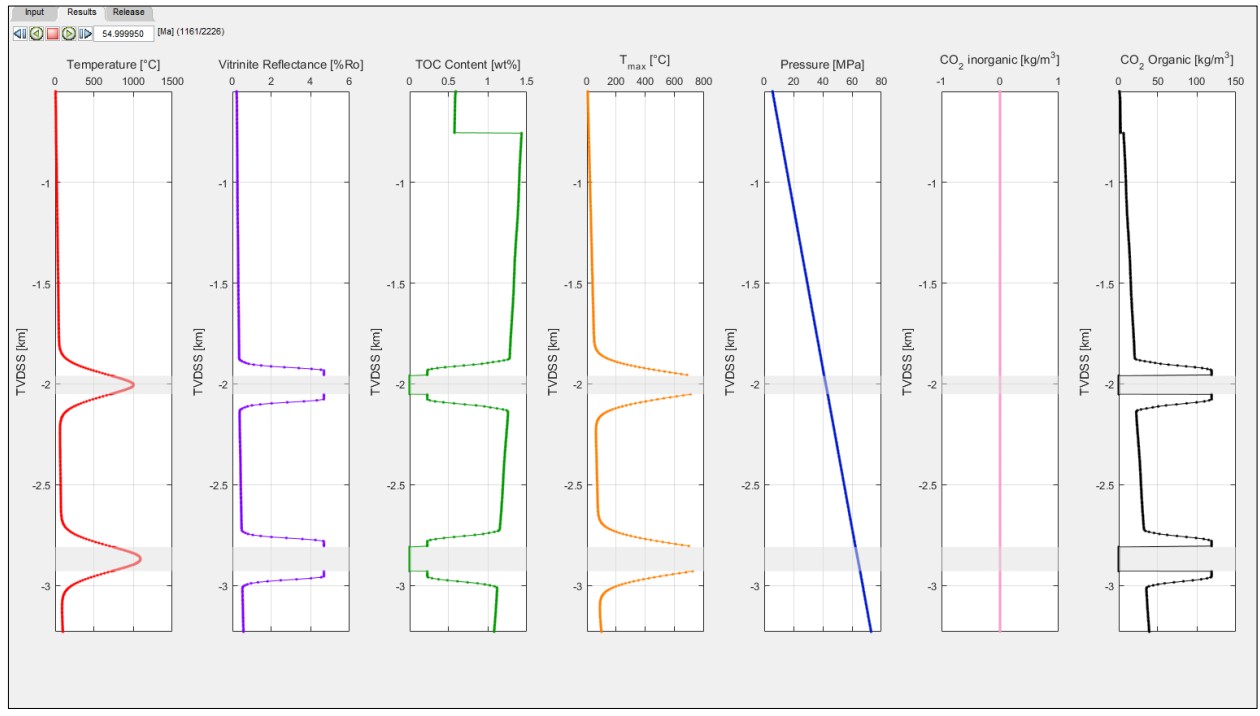


*Figure 6. Results tab 50 years after the emplacement of sills at 55 Ma for the Utgard High example. Sediments*

*around the sills are heated and $CO_2$ is liberated as organic matter is thermally degraded.*






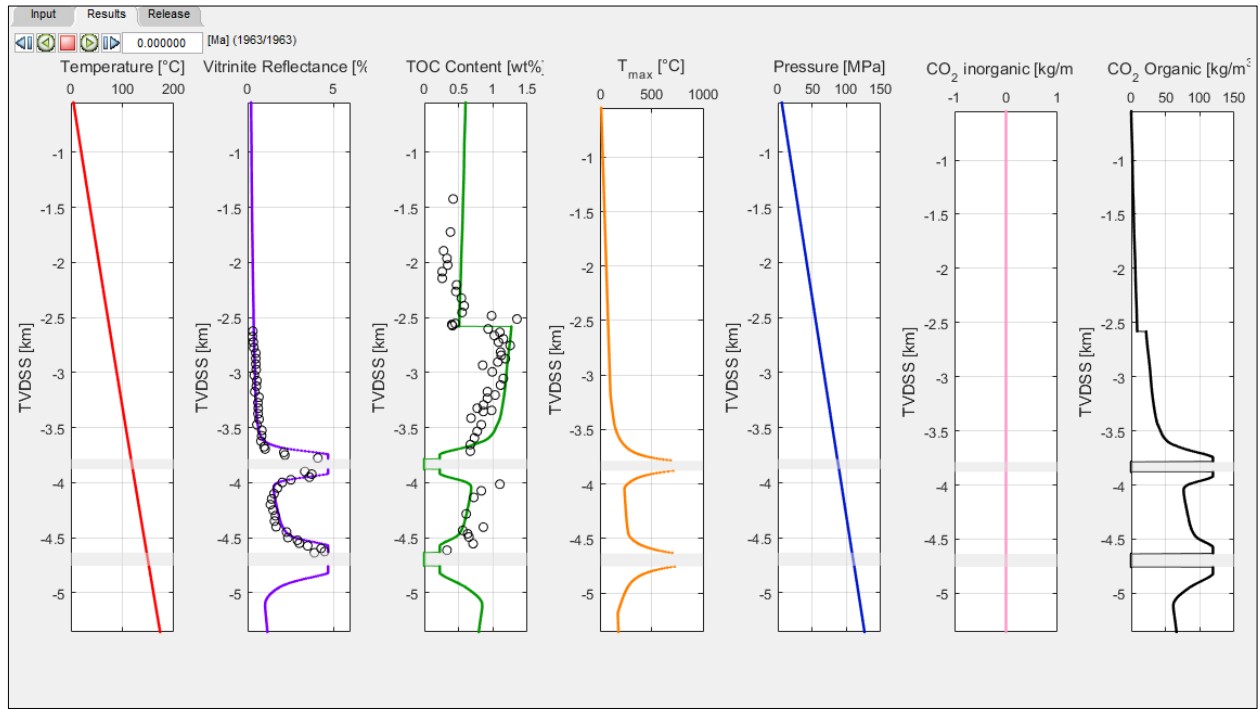

*Figure 7. Results tab at the end of simulation time for the Utgard High example. The present-day VR and TOC values (circles) show a good match with the model results.*

## 5.2 Example 2

The Karoo Large igneous province was emplaced through the Karoo Basin in South Africa in the Early Jurassic. The basin contains sills and dykes of varying thickness (Chevallier and Woodford, 1999; du Toit, 1920; Svensen et al., 2015; Walker and Poldervaart, 1949) , emplaced at about 182.6 Ma (Svensen et al., 2012). The basin stratigraphy consists of the Upper Carboniferous to the Triassic Karoo Supergroup and is divided in five groups (the Dwyka, Ecca, Beaufort, Stormberg and Drakensberg groups) with a postulated maximum cumulative thickness of 12 km and a preserved maximum thickness of 5.5 km (Tankard et al., 2009). The depositional environments of the sediments range from marine and glacial (the Dwyka Group), marine to deltaic (the Ecca Group), to fluvial (the Beaufort Group) and finally eolian (the Stormberg Group) (Catuneanu et al., 1998). The Karoo Basin is overlain by 1.65 km of preserved volcanic rocks of the Drakensberg Group, consisting mainly of stacked basalt flows erupted in a continental and dry environment (e.g., (Duncan et al., 1984)). Several recent





studies have been devoted to contact metamorphism of the organic-rich Ecca Group (Aarnes et al., 2011b;
Moorcroft and Tonnelier, 2016) and the possible consequences of thermogenic methane venting on the Early
Jurassic climate (Svensen et al., 2007; Svensen et al., 2015). Here we present two borehole cases from the
central (borehole KL1/78) and eastern (borehole LA1/68) parts of the basin previously studied and modelled by
Aarnes et al. (2011b) and Svensen et al. (2015), respectively. The details regarding the relative timing of sill
emplacement is poorly constrained and we thus use the same age for all sills. If the sills are closely spaced, this
will result in a higher maximum temperature in the sedimentary rocks between the sills (cf. (Aarnes et al.,
2011b)). For the erosion history of the Karoo Basin, we refer to Braun et al. (2014) and a rapid Late Cretaceous
erosion event.
**5.2.1 Karoo KL1/78**
The first example from the Karoo Basin is a short borehole with a length of 136 m that penetrates the Tierberg,
Whitehill and Prince Albert Formations. However, these Formations underlie a massive erosion sequence
consisting of 2.5 km of extrusives (Drakensberg Group) and 1.5 km of sediments (Stormberg and Beaufort
Groups) and are also included in the model. The borehole penetrates a single 15m thick sill at a depth of 72m
(Figure 10). The sill is emplaced within the Prince Albert Formation at 182.6 Ma at a temperature of 1150°C.
Initial average TOC data for the sedimentary layers is not known but can be roughly estimated using present-
day values. The initial TOC data is subsequently refined so that a better match of the model results to the
observed data is obtained, thereby highlighting how the model can be used to constrain initial conditions within
the sedimentary column (Figure 11). The importance of considering the entire basin history when constructing
the model is also emphasized by the VR results. The values of the VR results unaffected by the sill would be
much lower than the observed values if the eroded sequences are not considered. Addition of these layers to
the model results in added burial than would be expected than by just using the 136 m deep borehole. This
translates the VR curve laterally thereby better fitting the observed values (Figure 11). The final model shows a
good fit of TOC and VR to present day values. Model input data can be found in '1d_sill_input_kl178.xlsx'.



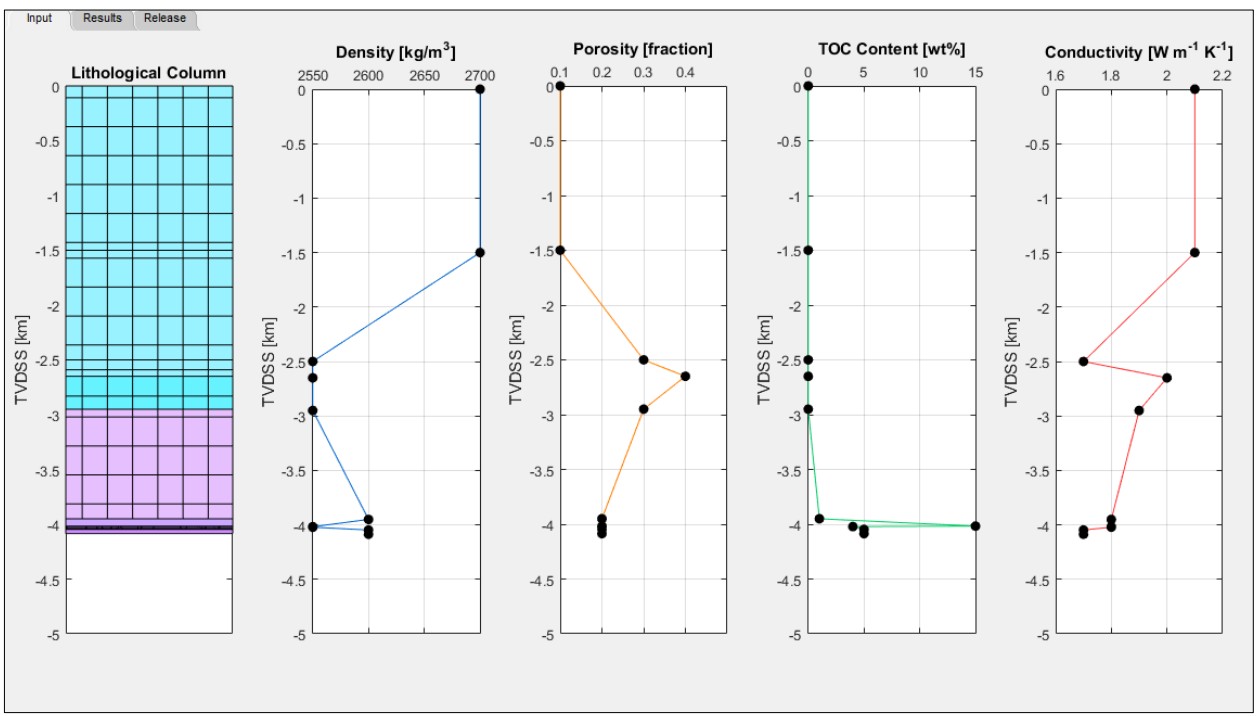


Figure 8. Input tab for KL178.


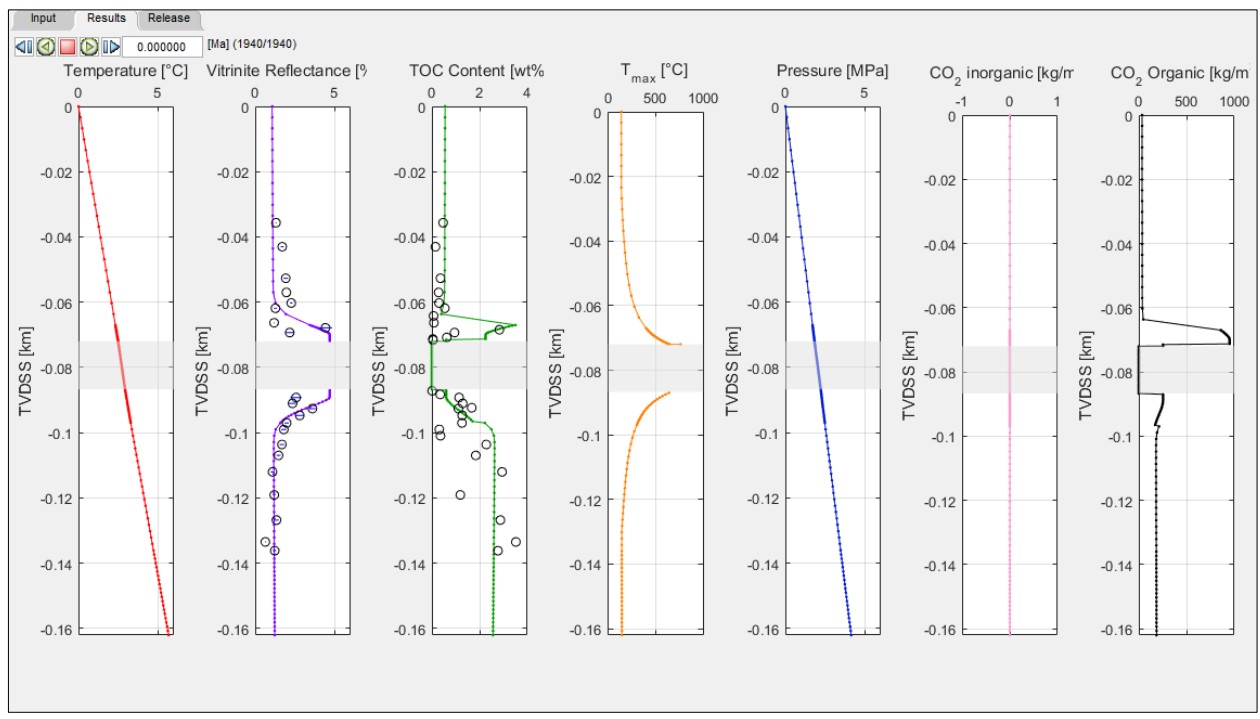




*Figure 9. Results tab at the end of simulation time for KL178 shows a good match to present-day TOC and VR values.*

### 5.2.2 Karoo LA1/68

The second example from the Karoo Basin is a borehole with a length of 1711 m that penetrates the basin down to the basement (Svensen et al., 2015). Additional erosional sequence consisting mostly of the Drakensberg lavas and a minor section of the Stormberg Group is also added. The borehole penetrates multiple sills throughout the entire column with thicknesses ranging from 2 to 132m (Figure 10). Initial average TOC data for the sedimentary layers is estimated from present-day values. Similar to the previous example, material properties are iteratively changed within realistic bounds to arrive at an initial setup that matches the final observations well (Figure 11). Model input data can be found in '1d_sill_input_la168.xlsx'.

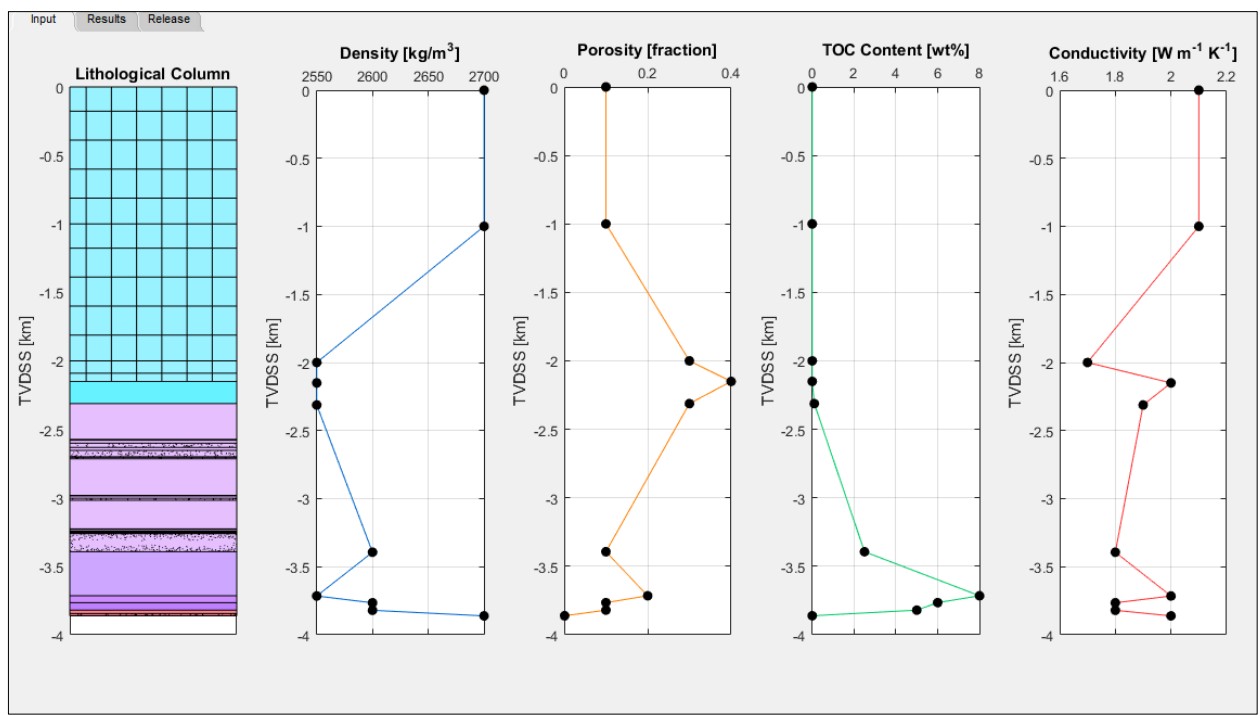

*Figure 10. Input tab for LA168.*





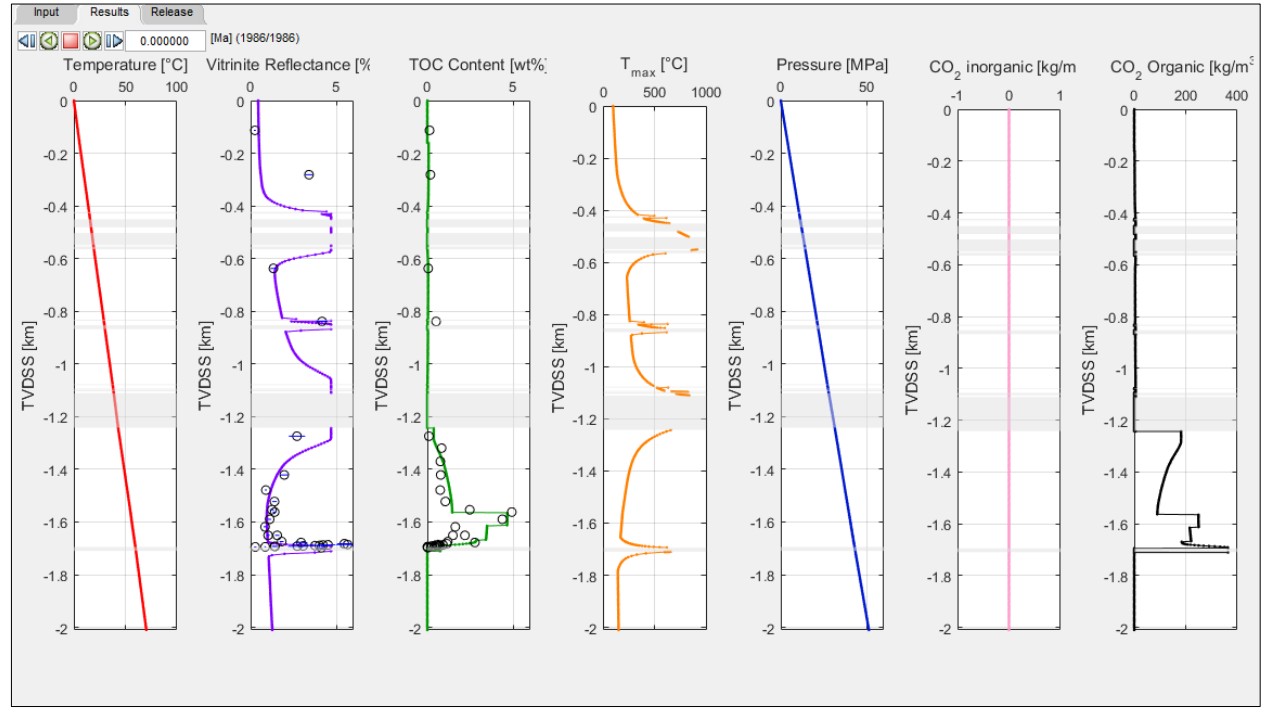

*Figure 11. Results tab at the end of simulation time for LA168 shows a good match to present-day TOC and VR values.*

# 6    Conclusions

- SILLi is a numerical model quantifies the thermal evolution of contact aureoles around sills emplaced in sedimentary basins. The model includes basin history (burial and erosion), thus providing background-maturation levels of organic matter and consequently more realistic gas production estimates.
- SILLi is a user-friendly tool that is written in Matlab and uses Excel for input data.
- The 1D tool allows for the quick quantification of the thermal effects of sill intrusions. The results can be, therefore, used to further constrain and test the initial conditions that may have been present within the lithological column that match present-day observations.
- Model output includes peak temperature profiles, post-metamorphic TOC content, vitrinite reflectivity, and the cumulative amount and rate of $CO_2$ generation. These values can be readily upscaled to basin



scales if the sill extent is known. The amount of $CO_2$ can also be easily converted to other carbon-bearing gases such as $CH_4$.

- Our three case studies demonstrate a good fit between aureole data (TOC and vitrinite reflectivity) and model output showing that the model can be successfully applied to basins in various global settings.

# 7  Code Availability and Software Requirements

The source code with examples is archived as a repository on Github/Zenodo (DOI: https://doi.org/10.5281/ZENODO.803748). Matlab 2014b or higher is required to run the code and Microsoft Excel or any equivalent software is required to edit .xls files.

# 8  License (BSD-2-Clause)

Copyright 2016 Karthik Iyer, Henrik Svensen and Daniel W. Schmid

Redistribution and use in source and binary forms, with or without modification, are permitted provided that the following conditions are met:

1. Redistributions of source code must retain the above copyright notice, this list of conditions and the following disclaimer.

2. Redistributions in binary form must reproduce the above copyright notice, this list of conditions and the following disclaimer in the documentation and/or other materials provided with the distribution.

THIS SOFTWARE IS PROVIDED BY THE COPYRIGHT HOLDERS AND CONTRIBUTORS "AS IS" AND ANY EXPRESS OR IMPLIED WARRANTIES, INCLUDING, BUT NOT LIMITED TO, THE IMPLIED WARRANTIES OF MERCHANTABILITY AND FITNESS FOR A PARTICULAR PURPOSE ARE DISCLAIMED. IN NO EVENT SHALL THE COPYRIGHT HOLDER OR CONTRIBUTORS BE LIABLE FOR ANY DIRECT, INDIRECT, INCIDENTAL, SPECIAL, EXEMPLARY, OR CONSEQUENTIAL DAMAGES (INCLUDING, BUT NOT LIMITED TO, PROCUREMENT OF SUBSTITUTE GOODS OR SERVICES; LOSS OF USE, DATA, OR PROFITS; OR BUSINESS INTERRUPTION) HOWEVER CAUSED AND ON ANY THEORY OF LIABILITY, WHETHER IN CONTRACT, STRICT LIABILITY, OR TORT (INCLUDING NEGLIGENCE OR OTHERWISE) ARISING IN ANY WAY OUT OF THE USE OF THIS SOFTWARE, EVEN IF ADVISED OF THE POSSIBILITY OF SUCH DAMAGE.



The software includes errorbarxy.m by Qi An (2016) (BSD-2-Clause License)
(http://www.mathworks.com/matlabcentral/fileexchange/40221).

## 9    Author Contributions

K. Iyer and D.W. Schmid developed the code. K. Iyer implemented the code and wrote the manuscript. H.
Svensen guided code development and provided input data from field studies. D. W. Schmid and H. Svensen
edited the manuscript.

## 10    Competing Interests

The authors declare that they have no conflict of interest.

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
