# Peer review of "SILLi 1.0: A 1D Numerical Tool Quantifying the Thermal Effects of Sill"

_Geoscientific Model Development, 2017_

## Referee Comment (RC1) · Anonymous Referee #1 · 4 Sep 2017

The authors present a good method for modeling the thermal effects of sill-like intrusions on host rocks. I recommend its publication in Geosci. Model Dev. Discussion after some minor revisions: 1) Please further highlight the novelty of SiLLi by comparing it with some other similar simulators such as MagmaHeatNS1D. MagmaHeatNS1D was developed based on almost the same models and written using an object-oriented language. In comparison, the Silli indeed considers some additional geological processes. Iver et al. needs to introduce the significance of these processes. Wang D., MagmaHeatNS1D: One-dimensional visualization numerical simulator for computing thermal evolution in a contact metamorphic aureole, Computers & Geosciences, 2013, 54(4): 21-27. 2) Line 153: modeling results are highly sensitive to boundary conditions. What kind of boundary condition is assumed for the upper and lower boundaries

by SiLLi? Besides, how to prove that "5 times the thickness of the bottommost sill" is reasonable? Such assumption needs to be made based on either special sensitivity analysis or the results of some similar researches. 3) Section 3.6: Iyer et al. consider some potential heat sink/source but ignored water boiling and vaporization. Why? For the one-dimensional thermal models, Jeager (1959), Barker et al. (1998, international journal of coal geology) Wang et al. (2007, GRL) and Wang (2011, international journal of coal geology) pointed out its effects on thermal evolution of host rocks. This may be explained in this section. 4) Section 3.6: although most organic-rich rocks are less permeable, Jaeger (1959), Galushkin (1997), Wang and Manga (2015) indeed showed the possible heat convection mechanism in shallowly buried shale host rocks. These work need to cited in this section.

—————————————————

---

## Referee Comment (RC2) · Anonymous Referee #2 · 10 Oct 2017

The manuscript SILLi 1.0: A 1D Numerical Tool Quantifying the Thermal Effects of Sill Intrusions by Iyer, et al is a welcome addition to the field that helps to provide a step forward in predictive modelling of intrusion effects on host rocks. The authors present a good method through the modeling with many sensible and clear outcomes of the 1D approach. I am pleased to recommend its publication in Geoscientific Model Development providing some minor revisions are addressed: 1. A little background and context to the modelling around intrusions would help the reader to see more clearly the novel aspects of this model (for example, perhaps some broader discussion early on as to the wider affects of intrusions on organic rich sedimentary successions, particularly with respect to hydrocarbon prospectivity (although not the primary focus of this paper will certainly be of significant interest to the field and indeed requires some finer background detail in line with the time spent on thermogenic gas, PETM etc). Suggested Refs to broaden background: Archer et al (2005); RodrÄś′guez et al (2005, 2006) – especially for comparison with the 2D models whithin, alongside some comparison of other modelling methods in refs already noted. Perhaps also Schofield et al 2015 or Muirhead et al 2017 for a broader view on organic matter alteration adjacent to intrusions). Specifically, why is this modelling method more applicable than/or add to the other modelling? 2. The correlation of modelled and actual TOC and VR is compelling, however the manuscript would benefit from some more detail about how the data is refined e.g. lines 384-386 and how this ties back to the methods above. Although TOC and VR have been typically used as a measurement around sills for decades – how does this correlate to other maturity parameters such as mineralogical markers, biomarkers? 3. The extent of the thermal aureoles of sills can be measured using TOC and VR (as discussed, among many other parameters). In the Model Input section these are displayed as 'optional'. Organic matter will frequently thermally alter in very different manners to mineralogical material and surely one or other parameter must be used to help gauge the full thermal impact of the sill? Clarity over the use of VR and/or TOC would help the reader. I look forward to seeing the modified manuscript and to any future work that develops this interesting, topical and essential model further.

---

## Author Response (AR1)

Dear Editor,

We have issued a point-by-point reply to all of the reviewers' comments in the open discussion on the geoscientific-model-development website. This also includes all of the changes made to the manuscript with respect to the comments.

Additionally, we have made some minor editorial changes to the manuscript (tracked in the manuscript) and minor enhancements to the model. A new release of the model has been uploaded to Github and has been assigned a Zenodo DOI (http://doi.org/10.5281/zenodo.1035878). The list of changes to the model are as follows:

Improvements:

- Conductivity in the energy equation is the geometric mean of fluid and rock conductivities.

- Result plotting during playback is significantly faster.

- Added an example of an igneous intrusion in a cooled pluton (i.e. no sedimentation or basin evolution involved).

Bug Fixes:

- Fixed plotting issue where in some cases whole range of observed data may not be plotted.

- Time step for last point in youngest sedimentary layer corrected.

- Fixed code so that file separators for Mac and Linux systems are recognized and used.

Best Regards,

Karthik Iyer

**Author's Response**

Dear Reviewer #1,

Thank you for your constructive review which helped us better evaluate the presented model and make suitable changes where required. A point-by-point answer to the review is as follows (line numbering according to revised manuscript):

1. Please further highlight the novelty of SiLLi by comparing it with some other similar simulators such as MagmaHeatNS1D. MagmaHeatNS1D was developed based on almost the same models and written using an object-oriented language. In comparison, the Silli indeed considers some additional geological processes. Iyer et al. needs to introduce the significance of these processes.

Wang D., MagmaHeatNS1D: One-dimensional visualization numerical simulator for computing thermal evolution in a contact metamorphic aureole, Computers & Geosciences, 2013, 54(4): 21-27.

- We have added the reference to Wang, 2013 in the introduction (Line 67) and also further highlighted the uniqueness of SILLi and the motivation behind the model (Lines 84-93: "The motivation behind the model and manuscript is to make a standardized numerical toolkit openly available that can be widely used by scientists with varying backgrounds to test the effect of magmatic bodies in a wide variety of settings using readily available data such as standard well logs and field measurements. The model incorporates relevant processes associated with heat transfer from magmatic intrusions such as latent heat effects, decarbonation reactions and organic matter maturation and also accounts for background maturation and erosion by systematically reconstructing the entire present-day sedimentary column from the input data. Lastly, the model results can be easily compared to the two most widely used aureole proxies in sedimentary rocks, vitrinite reflectance (VR) and total organic carbon (TOC) data").

2. Line 153: modeling results are highly sensitive to boundary conditions. What kind of boundary condition is assumed for the upper and lower boundaries by SiLLi? Besides, how to prove that "5 times the thickness of the bottommost sill" is reasonable? Such assumption needs to be made based on either special sensitivity analysis or the results of some similar researches.
   - The implementation of temperature boundary conditions to the upper and lower boundaries are already mentioned in Lines 179 to 181 ("The thermal solver computes the temperature within the deposited sedimentary column by applying fixed temperatures at the top and bottom at every step which are calculated from the prescribed geotherm"). We have added the reason and reference to justify that aureole processes are usually limited to less than 400% of sill thickness (Lines 173 to 176: "Note that the bottom boundary is extended to 5 times the thickness of the bottommost sill if that sill is close to or at the bottom boundary (hypothetical basement) in order to remove boundary effects and resolve aureole processes that are mostly limited to less than 4 times the sill thickness (Aarnes et al., 2010).").

3. Section 3.6: Iyer et al. consider some potential heat sink/source but ignored water boiling and vaporization. Why? For the one-dimensional thermal models, Jeager (1959), Barker et al. (1998, international journal of coal geology) Wang et al. (2007, GRL) and Wang (2011, international journal of coal geology) pointed out its effects on thermal evolution of host rocks. This may be explained in this section.
   - A full two-phase flow model would be required to fully capture the effect of pore water boing and subsequent condensation away from the heat source (e.g. Coumou et al., 2008. Phase separation, brine formation, and salinity variation at Black Smoker hydrothermal systems, JGR-Solid Earth). Moreover, previous studies have shown that model effects of the uncertainty of pre water volatilization is as large as the effects of variation in material properties such as heat capacity (Wang 2012. Comparable study on the effect of errors and uncertainties of heat transfer models on quantitative evaluation of thermal alteration in contact metamorphic aureoles: Thermophysical parameters, intrusion mechanism, pore-water volatilization and mathematical equations). Therefore, we have not implemented pore water volatilization in SILLi as it only adds to further uncertainty in an unconstrained variable.

4.  Section 3.6: although most organic-rich rocks are less permeable, Jaeger (1959), Galushkin (1997), Wang and Manga (2015) indeed showed the possible heat convection mechanism in shallowly buried shale host rocks. These work need to cited in this section.
    - The authors acknowledge that in some cases the effect of hydrothermal activity may indeed need to be considered in order to match field data as mentioned by the references above. The use of the Nusselt number approach (enhanced thermal conductivity) for such cases has been outlined in the manuscript (Lines 270 to 276: "Nevertheless, in some cases the effects of hydrothermal activity may be visible where the thermal aureole is larger above than below the sill and is recorded by vitrinite reflectance data (Galushkin, 1997; Wang and Manga, 2015). In such cases, the user may use an enhanced thermal conductivity (up to 5 times the usual rock conductivity) in the layer above the sill following the Nusselt number approach to account for hydrothermal activity and match field data. Note that care should be taken to check if the same effect can also be attributed to changes in other material properties or geological processes.").

Dear Reviewer #2,

Thank you for your constructive review which helped us better evaluate the presented model and make suitable changes where required. A point-by-point answer to the review is as follows (line numbering according to revised manuscript):

1.  A little background and context to the modelling around intrusions would help the reader to see more clearly the novel aspects of this model (for example, perhaps some broader discussion early on as to the wider affects of intrusions on organic rich sedimentary successions, particularly with respect to hydrocarbon prospectivity (although not the primary focus of this paper will certainly be of significant interest to the field and indeed requires some finer background detail in line with the time spent on thermogenic gas, PETM etc). Suggested Refs to broaden background: Archer et al (2005); Rodriguez et al (2005, 2006) – especially for comparison with the 2D models whithin, alongside some comparison of other modelling methods in refs already noted. Perhaps also Schofield et al 2015 or Muirhead et al 2017 for a broader view on organic matter alteration adjacent to intrusions). Specifically, why is this modelling method more applicable than/or add to the other modelling?
    - We have enhanced the introduction on modelling of sill intrusions by adding a short segment on the effects of intrusives on hydrocarbon prospectivity as suggested with the relevant references (Lines 52-59: "Magmatic intrusions are also of particular interest for hydrocarbon prospectivity and can impact petroleum systems in positive and negative ways (Archer et al., 2005; Monreal et al., 2009; Peace et al., 2017). High temperatures in the thermal aureole around such intrusions may induce maturation and hydrocarbon generation in immature, shallow strata that may have not been productive under normal burial. On the other hand, pre-emptive maturation of hydrocarbons around an intrusion may result in loss of hydrocarbons if a suitable reservoir has not yet formed. Additionally, pre-existing oil in a reservoir may crack to gas in the vicinity of magmatic intrusions resulting in degradation of a potential prospect."). We have also modified the introduction so that it better conveys the motivation behind the model and its strengths (Lines 84-93: "The motivation behind the model and manuscript is to make a standardized numerical toolkit openly available that can be widely used by scientists with varying backgrounds to test the effect of magmatic bodies in a wide variety of settings using readily available data such as standard well logs and field measurements. The model incorporates relevant processes associated with heat transfer from magmatic intrusions such as latent heat effects, decarbonation reactions and organic matter maturation and also accounts for background maturation and erosion by systematically reconstructing the entire present-day sedimentary column from the input data. Lastly, the model results can be easily compared to the two most widely used aureole proxies in sedimentary rocks, vitrinite reflectance (VR) and total organic carbon (TOC) data.").

2. The correlation of modelled and actual TOC and VR is compelling, however the manuscript would benefit from some more detail about how the data is refined e.g. lines 384-386 and how this ties back to the methods above. Although TOC and VR have been typically used as a measurement around sills for decades – how does this correlate to other maturity parameters such as mineralogical markers, biomarkers?

   - We have elaborated on how we refine the input TOC data in the manuscript if this data is not known (Lines 424-429: "Initial average TOC data for the sedimentary layers away from the sill intrusion is not known but can be roughly estimated using present-day values, i.e. the TOC values will be higher than current values as TOC is thermally broken down close to the intrusion. The initial input TOC data is subsequently refined so that a better match of the model results to the observed data is obtained, thereby highlighting how the model can be used to constrain initial conditions within the sedimentary column (Figure 9)."). In order to do so, we first estimate the TOC content, if it is not measured, by using a higher than present day value since TOC is thermally broken down with time. If the values do not result in a good fit to present-day TOC values than the initial input values are subsequently refined until a good fit is obtained. We have also stated that the user can correlate other, 'non-standard' maturity parameters themselves using its correlation to the temperature-time evolution from the model results if this correlation is known (Lines 280-282: "Similarly, correlation to other maturity parameters such as mineralogical markers or biomarkers (e.g. Muirhead et al. (2017)) can be performed by the user using the time-temperature evolution from the model if so desired.").

3. The extent of the thermal aureoles of sills can be measured using TOC and VR (as discussed, among many other parameters). In the Model Input section these are displayed as 'optional'. Organic matter will frequently thermally alter in very different manners to mineralogical material and surely one or other parameter must be used to help gauge the full thermal impact of the sill? Clarity over the use of VR and/or TOC would help the reader.

- Present-day measured TOC and VR values are 'optional' in terms of user-input to the model since these values are not always measured or available. This does not inhibit the user from running the model.

[revised manuscript text omitted]